# Intrinsic disorder within an AKAP-protein kinase A complex guides local substrate phosphorylation

F Donelson Smith[1†], Steve L Reichow[2†], Jessica L Esseltine[1], Dan Shi[2], Lorene K Langeberg[1], John D Scott[1]*, Tamir Gonen[2]*

[1]Department of Pharmacology, Howard Hughes Medical Institute, University of Washington, Seattle, United States; [2]Janelia Farm Research Campus, Howard Hughes Medical Institute, Ashburn, United States

**Abstract** Anchoring proteins sequester kinases with their substrates to locally disseminate intracellular signals and avert indiscriminate transmission of these responses throughout the cell. Mechanistic understanding of this process is hampered by limited structural information on these macromolecular complexes. A-kinase anchoring proteins (AKAPs) spatially constrain phosphorylation by cAMP-dependent protein kinases (PKA). Electron microscopy and three-dimensional reconstructions of type-II PKA-AKAP18γ complexes reveal hetero-pentameric assemblies that adopt a range of flexible tripartite configurations. Intrinsically disordered regions within each PKA regulatory subunit impart the molecular plasticity that affords an ~16 nanometer radius of motion to the associated catalytic subunits. Manipulating flexibility within the PKA holoenzyme augmented basal and cAMP responsive phosphorylation of AKAP-associated substrates. Cell-based analyses suggest that the catalytic subunit remains within type-II PKA-AKAP18γ complexes upon cAMP elevation. We propose that the dynamic movement of kinase sub-structures, in concert with the static AKAP-regulatory subunit interface, generates a solid-state signaling microenvironment for substrate phosphorylation.

*For correspondence:
scottjdw@u.washington.edu
(JDS); gonent@janelia.hhmi.org
(TG)

†These authors contributed equally to this work

Competing interests: The authors declare that no competing interests exist.

## Introduction

Intrinsically disordered regions of proteins are widespread in nature, yet the mechanistic roles they play in biology are underappreciated. Such disordered segments can act simply to link functionally coupled structural domains or they can orchestrate enzymatic reactions through a variety of allosteric mechanisms (*Dyson and Wright, 2005*). The regulatory subunits of protein kinase A provide an example of this important phenomenon where functionally defined and structurally conserved domains are connected by intrinsically disordered regions of defined length with limited sequence identity (*Scott et al., 1987*). In this study, we show that this seemingly paradoxical amalgam of order and disorder permits fine-tuning of local protein phosphorylation events.

Phosphorylation of proteins is a universal means of intracellular communication that is tightly controlled within the spatial context of the cell. A variety of stimuli trigger these events, which are catalyzed by numerous protein kinases and reversed by phosphoprotein phosphatases (*Hunter, 1995*). A classic example is production of the second messenger cyclic AMP (cAMP), which stimulates a cAMP-dependent protein kinase (PKA) to phosphorylate a range of cellular targets (*Taylor et al., 2012*). The PKA holoenzyme is a tetramer composed of two regulatory subunits (R) and two autoinhibited catalytic subunits (PKAc). Binding of cAMP to each R subunit is believed to liberate active kinase and phosphorylation ensues. The local action of PKA is dictated by A-kinase anchoring proteins (AKAPs) that impose spatial constraint by tethering this kinase in proximity to substrates (*Wong and Scott,*

**eLife digest** It was once thought that proteins needed to have structures that were both ordered and stable, but this view was changed by the discovery that certain proteins contain regions that are disordered and flexible. In some cases these regions of intrinsic disorder help the protein to function by linking more stable regions that are active. However, in other proteins the disordered regions are themselves biologically active and can, for example, function as enzymes.

Protein kinase A is a family of enzymes that contains both ordered and disordered regions, with the ordered sections being involved in phosphorylation, a chemical process that is widely used for communication within cells. However, in order to initiate phosphorylation, these kinases must be anchored to a rigid substrate nearby, so a second group of proteins called AKAPs–which is short for A-kinase anchoring proteins–hold the kinases in place by binding to their disordered regions. These AKAPs also help the kinases to dock with other molecules involved in phosphorylation.

A full structural picture of how the kinases induce phosphorylation has yet to be obtained, partly because it is extremely difficult to determine the structure of the disordered regions within the kinases. Moreover, the AKAPs are also disordered, which makes it difficult to work out how the kinases are held in position.

Smith, Reichow *et al.* have used electron microscopy to reveal that the disordered region has two important roles: it determines how far away from the anchoring protein that the active region of the kinase can operate, and it influences how efficiently the kinase can bind to its target molecule in order to induce phosphorylation. Future challenges include investigating how the inherent flexibility of AKAP complexes contribute to the efficient phosphorylation of physiological targets.

*2004*). AKAPs also organize higher-order macromolecular signaling complexes through their association with G-protein coupled receptors, GTPases and additional protein kinases. Likewise, AKAP-associated phosphatases and phosphodiesterases act to locally terminate these signals. While physiological roles for AKAPs that sequester enzymatic activity with ion channels, cytoskeletal components and regulatory enzymes have been well established, the structural mechanisms involved in these protein–protein interactions have been difficult to characterize.

Currently, structural details on PKA anchoring are limited because most AKAPs are large, intrinsically disordered macromolecules that lack recognizable structural domains. An exception is the crystal structure of the central domain of AKAP18γ that bears homology to bacterial 2H phosphoesterase domains (*Gold et al., 2008*). Likewise, high-resolution crystallographic structures of the catalytic subunit (PKAc) when free and in complex with the C-terminal autoinhibitory and cAMP binding domains of the type I or type II regulatory subunits of PKA (RI and RII) have provided details on the mechanisms of catalysis and autoinhibition (*Knighton et al., 1991*, *1992*; *Gold et al., 2006*, *2008*; *Wu et al., 2007*). Yet, despite decades of effort, a complete structural picture of the PKA holoenzyme is lacking. This is presumably due to the presence of long flexible intrinsically disordered linker regions within the R subunit that tether this complex. NMR spectroscopy and X-ray crystallographic studies show that the N-terminal domains of RI and RII homodimerize through a four-helix bundle docking and dimerization interface (D/D) (*Newlon et al., 2001*; *Gold et al., 2006*; *Kinderman et al., 2006*; *Sarma et al., 2010*). The D/D creates a high-affinity binding groove for a canonical amphipathic helix on each AKAP that forms the reciprocal binding surface (*Gold et al., 2006*).

Although the physiological consequences of anchored PKA phosphorylation events have been established in a variety of cellular contexts, we have yet to discern how the individual protein components are assembled and operate within AKAP complexes (*Scott and Pawson, 2009*). Even more elusive is a mechanistic role for intrinsically disordered domains within these macromolecular assemblies. For example, does internal flexibility within these signaling assemblies modulate other aspects of enzyme action in addition to sequestering PKA at specific regions of the cell? Here we have used electron microscopy (EM) to evaluate the topological arrangement of the fully assembled type IIα PKA holoenzyme when anchored to AKAP18γ. This analysis unveils a remarkable level of conformational plasticity that resides within the AKAP–PKA complex. In vitro and cell-based structure-function approaches reveal an unexpected functional requirement for intrinsically disordered regions within RIIα. These regions not only define a radius of action of the anchored catalytic subunit but also modulate the enzymatic efficiency

of the kinase. Insights from these studies may be broadly applicable to the understanding of other intrinsically disordered proteins and anchoring complexes that organize enzymatic action in the cell.

## Results and discussion

Macromolecular assemblies were formed when purified human γ isoform of AKAP18 (AKAP18γ) was incubated with the PKA holoenzyme (PKA$_{holo}$), formed by the RIIα and PKAc subunits (*Figure 1A*, 'Materials and methods'). Following separation by size-exclusion chromatography, fractions from the front of the elution peak (indicated by arrow in *Figure 1B*) were analyzed by SDS-PAGE (*Figure 1C*, left). The subunit composition was confirmed by immunoblotting (*Figure 1C*, mid). Native electrophoresis established that a majority of this material migrated as a single species with an apparent molecular weight in excess of 240 kD (*Figure 1C*, right). This molecular mass is consistent with a hetero-pentameric complex composed of a single AKAP18γ molecule anchored to an RIIα subunit dimer and two PKAc subunits.

The structure of the AKAP18γ–PKA$_{holo}$ complex was resolved by electron microscopy of negatively stained particles since these complexes are too small to be imaged in vitrified ice (*Figure 1D*). Single particle analysis revealed clusters of three densities, each approximately 60–100 Å in size resembling beads on a string (*Figure 1D,E*). Closer inspection of individual particles established that these tripartite structures adopted a range of conformations (*Figure 1E*). In solution, these complexes may adopt many more configurations and flattening on the carbon support of the EM grid likely captured only a subset of the possible topologies. Approximately 7,000 particles were selected from electron micrographs for structural analysis. Projection averages were derived by classifying particles of similar orientation and structural conformation using reference-free multivariate statistical analysis (*van Heel et al., 1996*), and iterative stable alignment and clustering procedures (*Yang et al., 2012*) (*Figure 1F*, 'Materials and methods'). Class averages revealed a remarkable variety of configurations (*Figure 1F*; *Video 1*). These ranged from a tightly packed pseudo-symmetric triangular configuration (*Figure 1F*, left panel) to a fully extended linear configuration of the three densities (*Figure 1F*, right panel). In either case, the central density was consistently smaller (~60 Å) than the two peripheral densities (85 × 100 Å). Similar size differences between the central and peripheral lobes were observed at the single particle level in raw micrographs (*Figure 1D,E*). Affinity-labeling of strep-tagged AKAP18γ with a 5 nm gold particle targeted the smaller central density, indicating that this element includes the anchoring protein (*Figure 1D*, inset).

Three-dimensional (3D) reconstructions for the triangular and linear configurations of the AKAP18γ–PKA$_{holo}$ complex were determined at 35 Å resolution from a tilted-series dataset (*Figure 2A*, *Figure 2—figure supplements 1 and 2*). Negative staining in EM can cause flattening of particles, which might lead to apparent structural distortions. However, inspection of particles at various tilt angles showed that particles of linear and triangular conformations remained clearly distinguishable, even at high tilt angles (*Figure 2—figure supplement 1*). In both 3D reconstructions, a central mass with dimensions of 60 × 60 × 80 Å (corresponding to the site of AKAP18γ as demonstrated by affinity gold labeling, *Figure 2A*, black triangle and *Figure 1D*, inset) was flanked on either side by larger densities of 100 × 100 × 85 Å (*Figure 2A*). These flanking densities can each accommodate a sub-complex of RIIα and PKAc. In the triangular conformation, the two peripheral densities are oriented at a 100° angle with respect to the central density and exhibit an end-to-end length of ~300 Å (*Figure 2A*, top). The end-to-end length increases to ~385 Å in the extended linear configuration (*Figure 2A*, bottom). Back-projections calculated from the final 3D maps compare well with the experimental class averages (*Figure 2—figure supplement 2*). Moreover, this back-projection analysis demonstrated that when the triangular model is tilted completely on its edge (where it may appear more linear in projection) its dimensions (max length = 300 Å) are significantly smaller than the maximum end-to-end length obtained for the linear reconstruction (max length = 385 Å). Hence, we conclude that the linear and triangular conformations are structurally distinct.

Pseudo-atomic models of the pentameric protein assembly in both configurations were constructed by fitting Protein Data Bank coordinates for regions of AKAP18γ and PKA$_{holo}$ subunits (*Knighton et al., 1991*, *1992*; *Gold et al., 2006*, *2008*; *Wu et al., 2007*) (*Figure 2A*, 'Materials and methods'). A model for AKAP18γ (residues 88–317) was derived by connecting the central domain (residues 88–290) (*Gold et al., 2008*) via a short linker to the PKA anchoring helix (AKAP$_{(helix)}$, residues 301–317) (*Gold et al., 2006*) (*Figure 2A*, yellow , 'Materials and methods'). When AKAP18γ is docked to residues 1–43 of RIIα (RII$_{(D/D)}$), the resulting sub-structure fits within the central density of maps for both the triangular and extended configurations (*Figure 2A*, yellow). This tallies with the single particle affinity-labeling

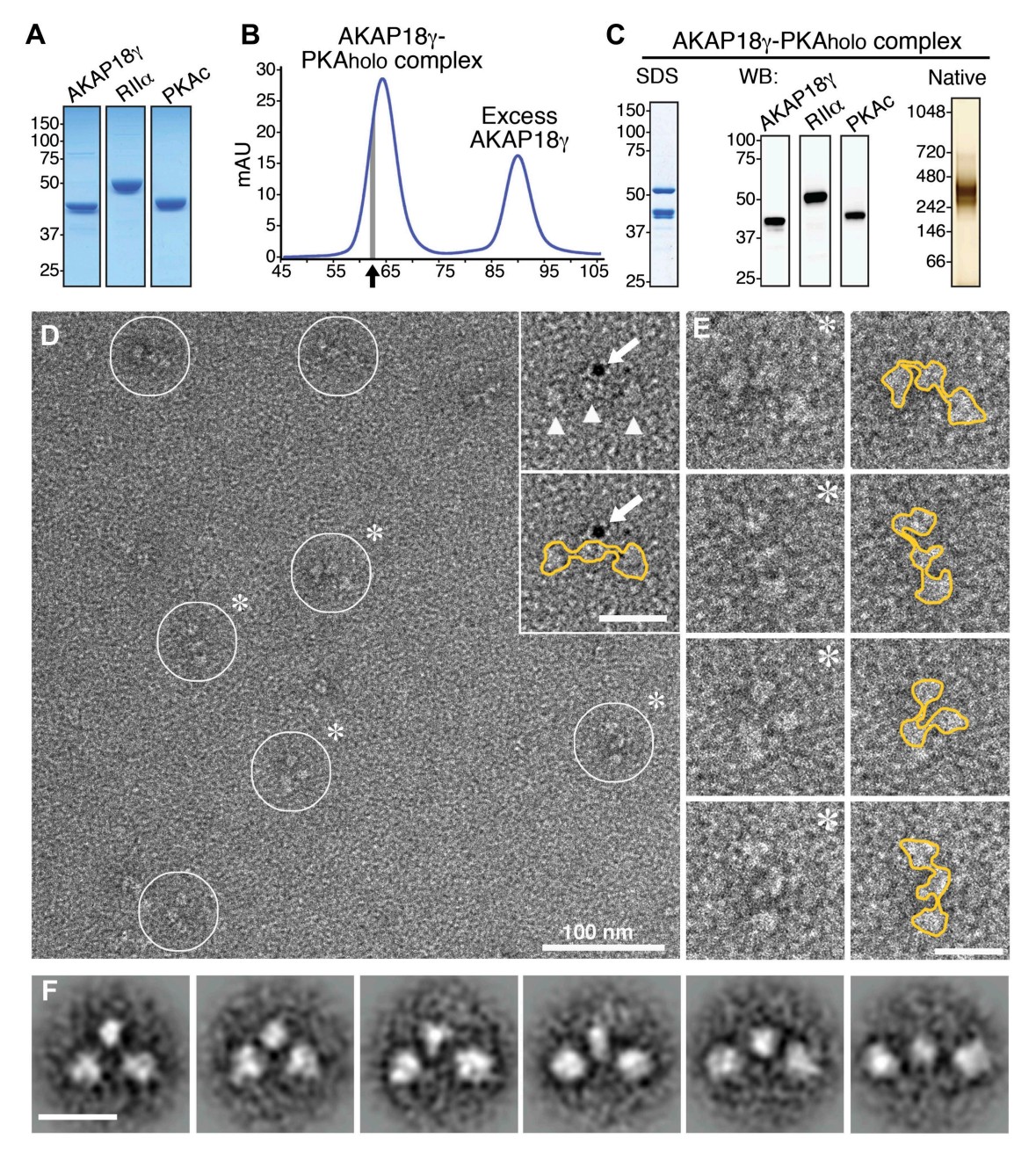

**Figure 1**. Purification and electron microscopy of the AKAP18γ–PKA$_{holo}$ complex. (**A**) SDS-PAGE and Coomassie staining of purified individual complex components. (**B**) Size-exclusion chromatography (SEC) trace for purification of the assembled AKAP18γ–PKA$_{holo}$ complex. Fractions at the leading edge of the peak (indicated by gray bar) were chosen for further analysis. (**C**) SDS-PAGE (left), western blot (middle) and native gel electrophoresis (right) obtained from the SEC peak elution fraction (arrow in **B**). (**D**) Electron micrograph of the negatively stained AKAP18γ–PKA$_{holo}$ complexes (circles). Triangles indicate the three major densities of the AKAP18γ–PKA$_{holo}$ complex. Inset, shows labeling with a gold nanoparticle (arrow) conjugated to an AKAP18γ-streptavidin moiety (arrow). (**E**) Left, enlarged images of individual AKAP18γ–PKA$_{holo}$ complexes (denoted by asterisks in **D**). (**E**) Right, highlighted outline (yellow) of particle shapes. (**F**) Projection averages of the AKAP18γ–PKA$_{holo}$ complex classified into distinct conformations using ISAC (*Yang et al., 2012*). Unlabeled scale bars represent 25 nm.

studies that map AKAP18γ to this region (*Figure 1D*, inset). In a similar manner, the peripheral densities each accommodate a PKA sub-structure consisting of one catalytic subunit (*Figure 2A*, blue) in complex with the cAMP-binding domain of RIIα, residues 91–392 (*Figure 2A*, green) (*Knighton et al., 1991*; *Wu et al., 2007*). Finally these models were completed by connecting the central

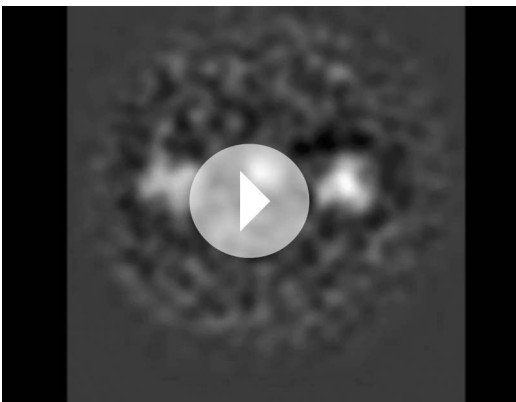

**Video 1**. Conformational dynamics of the wild-type AKAP18γ–PKA holoenzyme complex. A montage of projection averages obtained for the wild-type AKAP18γ–PKA holoenzyme complex displays the variety of topological configurations sampled by the dynamic signaling particle.

AKAP18γ–RII$_{(D/D)}$ sub-complex to each RIIα-PKAc unit through a flexible chain corresponding to residues 44–90 of RIIα (*Figure 2A*, 'Materials and methods'). The complete pseudo-atomic model of AKAP18γ–PKA$_{holo}$ complex is presented in *Figure 2B*.

Our model of the anchored PKA complex implies that an intrinsically disordered flexible linker region within RIIα supports the array of conformations that were observed in the raw micrographs and the projection averages (*Figures 1 and 2C*). To quantitatively assess the distribution of conformations assumed by this complex, we measured the end-to-end distance between the two large peripheral densities of 223 individual particles (*Figure 2D*). This population of structures followed a Gaussian distribution (*Figure 2D*, green trace) with a mean particle length of 275 ± 65 Å (n = 223). We propose that conformational plasticity observed in these analyses is facilitated by this intrinsically disordered region between residues 44 and 90 of RIIα, a linker that connects the AKAP docking site (D/D domain) to the cAMP-responsive transduction domains. This notion is further substantiated by a primary sequence analysis of RIIα orthologs, showing that the linker regions are of similar length but exhibit low amino acid identity (*Figure 3A*).

We reasoned that if the linker region in RIIα contributes to conformational flexibility of the holoenzyme, altering its length could affect the structure and function of the anchored kinase. This structural postulate was tested by producing modified AKAP18γ–PKA$_{holo}$ complexes in which the linker region of RIIα was deleted (RIIα Δ44–86) or replaced with an extended sequence of 60 residues found in zebrafish (RIIα ZeChimera; *Figure 3A,B*). Formation of modified AKAP–PKA complexes (assembled as described above) was monitored by Coomassie blue staining and immunoblot detection of the component proteins (*Figure 3C*). The conformations of these modified AKAP18γ–PKA$_{holo}$ complexes were analyzed by electron microscopy as previously described (*Figure 3D–G*). Single-particle EM and class averages of the assemblies formed with RIIα Δ44–86 yielded uniform complexes exclusively in a compact triangular configuration (*Figure 3D,F*). In contrast, complexes formed with RIIα ZeChimera resembled the array of conformations observed for the wild-type AKAP18γ–PKA$_{holo}$ assembly (*Figure 3E,G*). Quantitative analysis of the differing linker lengths was assessed by measuring the radius of each particle, defined by the center of the AKAP18γ subunit to the distal end of each PKA subunit. Particle radii for the RIIα Δ44–86 complexes ranged in length from 55 to 125 Å, (mean value of 87 ± 13 Å, n = 142; *Figure 3H*) whereas the assemblies formed with the RIIα ZeChimera were extended with lengths ranging from 100 to 265 Å (mean value of 168 ± 33 Å, n = 296; *Figure 3H*). These latter measurements are similar to the parameters of the native complex, (160 ± 29 Å, n = 216; *Figures 2D and 3D*). Thus we conclude that a flexible linker in RIIα is responsible for the conformational plasticity of the AKAP18γ–PKA$_{holo}$ assemblies.

One mechanistic ramification of our structural analyses is that flexibility within PKA$_{holo}$ complex could permit precise orientation of the anchored catalytic subunit toward substrates. This would be particularly true for substrates that are physically associated with AKAPs. For example, the type 4 phosphodiesterase isoforms PDE4D3 and PDE4D5 associate with the central domain of AKAP18γ and are phosphorylated on two sites by PKA (*Sette and Conti, 1996*; *Carlisle Michel et al., 2004*; *Stefan et al., 2007*). We confirmed this protein–protein interaction upon co-expression of the components in HEK293 cells (*Figure 4A*). PDE4Ds co-precipitate with AKAP18γ and the RII subunits of PKA (*Figure 4A*, lane 1) but not with GFP controls (*Figure 4A*, lane 2). Accordingly, cAMP phosphodiesterase activity was enriched 3.27 ± 0.48-fold (n = 5) in AKAP18γ–GFP immune complexes as compared to GFP controls or samples treated with the PDE4 selective inhibitor rolipram (*Figure 4B*). These data allowed us to move to an in vitro system to study phosphorylation of PDE4D by AKAP18–PKA complexes.

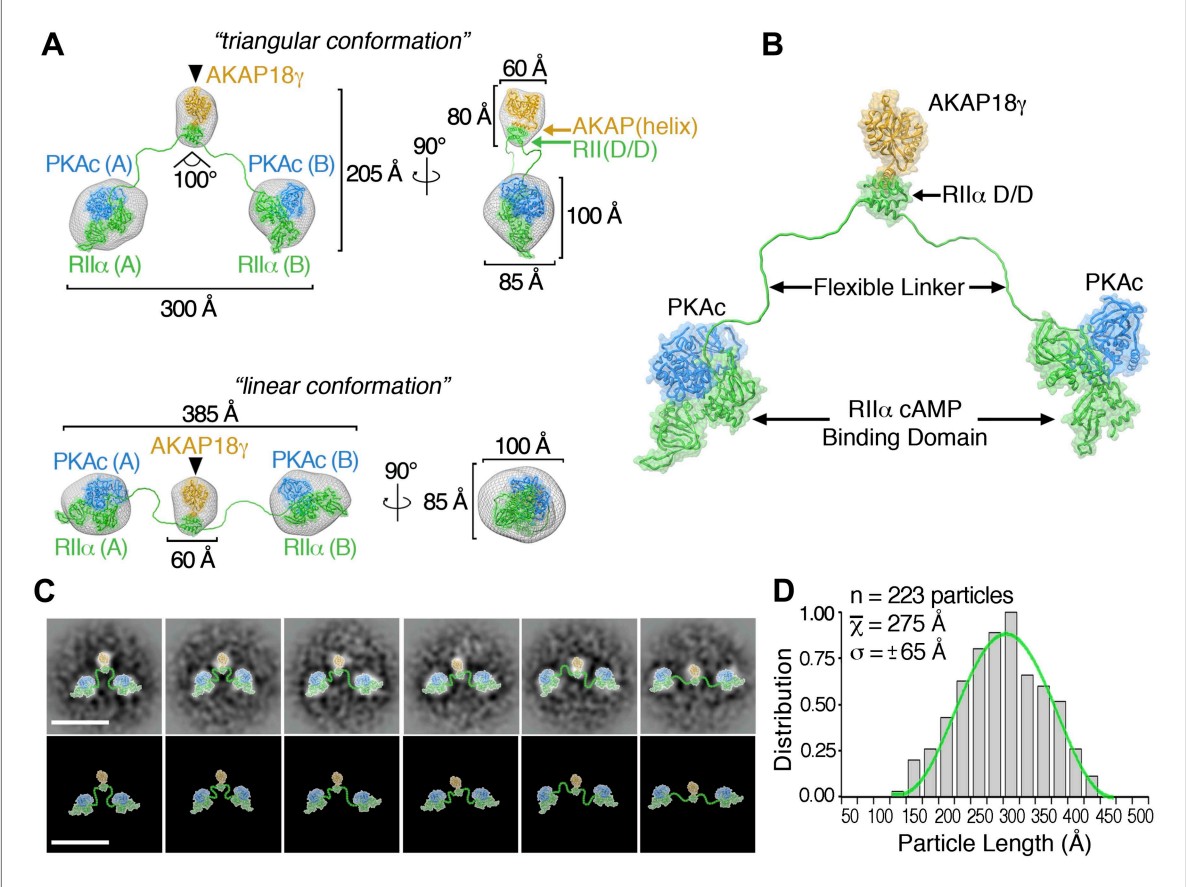

**Figure 2**. 3D reconstructions and pseudo-atomic structure of the AKAP18γ–PKA$_{holo}$ complex. (**A**) Three-dimensional (3D) EM reconstructions and 90° rotated views of the fitted molecular models for the AKAP18γ–PKA$_{holo}$ complex. High-resolution structures for regions of AKAP18γ (gold), RIIα (green) and C subunit of PKA (blue) that fit within the EM densities are indicated. Models are presented in the (top) compact triangular and (bottom) extended linear conformation. (**B**) Pseudo-atomic model of the AKAP18γ–PKA$_{holo}$ complex. (**C**) (top) Projection averages of the AKAP18γ–PKA$_{holo}$ complex with structural domains fitted into the EM densities and connected by lines representing the RIIα flexible linker regions. (bottom) Projection averages were removed for clarity. Scale bar represents 25 nm. (**D**) Statistical analysis of individual particle lengths in angstroms (Å) displays a Gaussian distribution (green line) with a mean value of 275 Å and a standard deviation (σ) ±65 Å.

The following figure supplements are available for figure 2:

**Figure supplement 1**. Tilted-series electron microscopy data.

**Figure supplement 2**. Three-dimensional EM maps of the AKAP18γ–PKA$_{holo}$ complex.

In vitro phosphorylation studies on AKAP18-associated PDE4D were performed in two phases. Our structural data predict that the C subunit of anchored PKA resides within ~100 Å of its substrate PDE4D (shown schematically in *Figure 4C*). We reasoned that this tight configuration could permit cAMP-independent phosphorylation of the phosphodiesterase. Therefore, in the first phase, 'basal' phosphorylation of PDE4D was measured in the context of the AKAP18γ signaling complex (*Figure 4D*). Phosphate incorporation into PDE4D was increased 1.87 ± 0.14-fold (p<0.05) upon its tethering to AKAP18γ as assessed by autoradiography (*Figure 4D, E*). This suggests that the proximity to a substrate afforded by AKAP18γ augments the cAMP-independent action of PKA. This could explain why physiologically relevant PKA targets such as aquaporins or phospholamban, which require continual or instantaneous phosphorylation to fulfill their biological roles, have their own AKAP-associated pool of kinase (*Henn et al., 2004*; *Lygren et al., 2007*; *Gold et al., 2012*). Accordingly, a primary function of AKAPs may be to ensure basal phosphorylation of such tethered substrates.

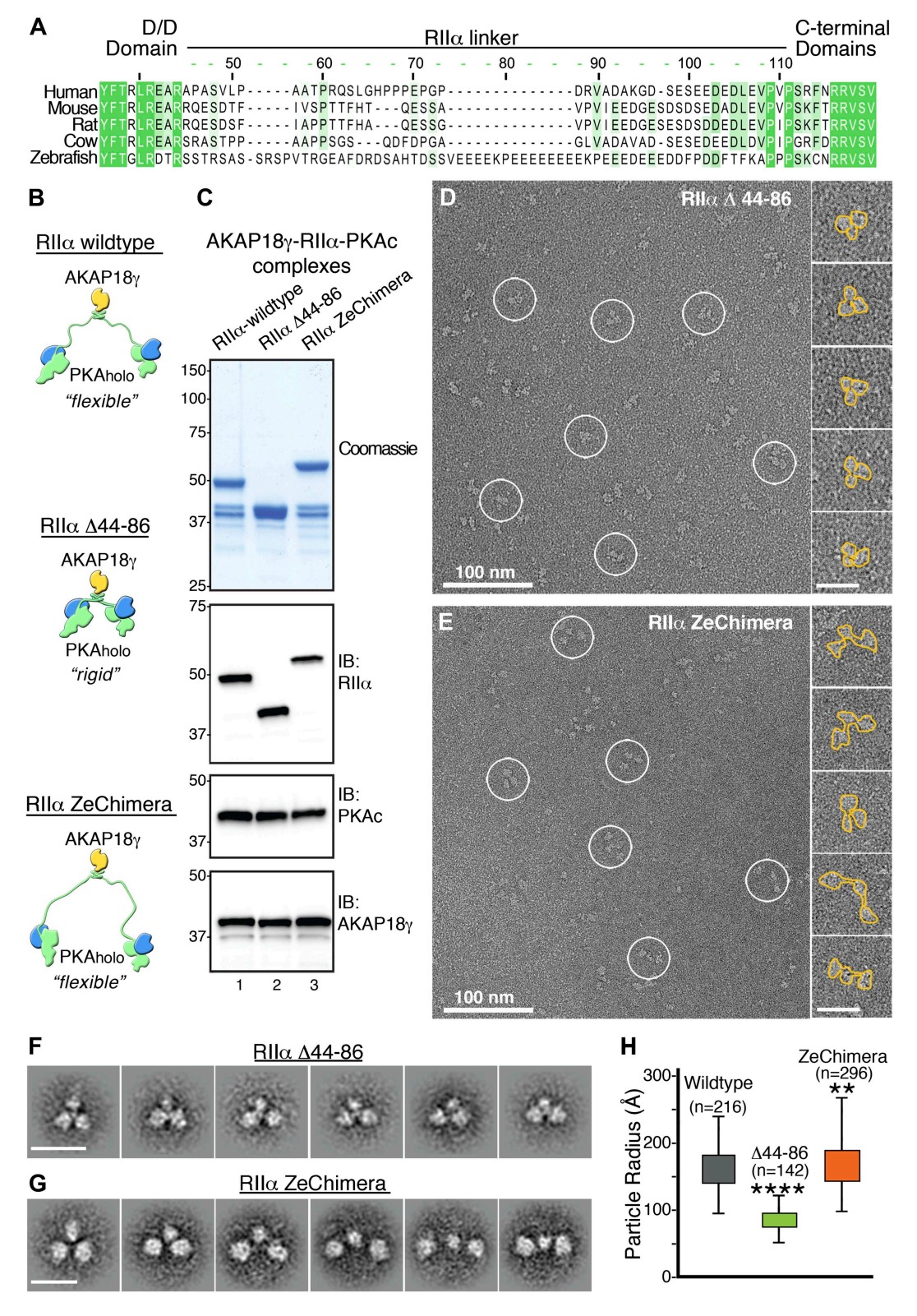

**Figure 3.** Flexibility within RIIα constrains the configuration of the anchored kinase assembly. (**A**) Amino acid sequence alignment of the linker region in RIIα that connects the conserved N-terminal D/D domain to the C-terminal autoinhibitor and cAMP binding domains. This region shows low sequence homology and is likely to be structurally disordered. (**B**) Schematic representations of modified AKAP18γ–PKA_holo complexes with AKAP18γ depicted in *Figure 3. Continued on next page*

Figure 3. Continued

yellow, RIIα in green and PKAc in blue. (**C**) Biochemical analysis of the purified AKAP18γ–RIIα-PKAc complexes assembled with the wild-type RIIα subunit, the RIIα Δ44–86 mutant where the linker region was deleted, and the RIIα ZeChimera mutant where the mouse RIIα linker region was replaced with the corresponding and extended sequence from zebrafish. Top panel shows SDS-PAGE and Coomassie blue staining of the protein components. The next three panels show western blotting for RIIα, PKAc subunit and AKAP18γ, respectively. (**D**) Electron micrograph of negatively stained AKAP18γ–PKA$_{holo}$ complexes (circles) assembled with the RIIα Δ44–86 construct. (**E**) Electron micrograph of negatively stained AKAP18γ–PKA$_{holo}$ complexes (circles) assembled with the RIIα ZeChimera construct. Insets in (**D** and **E**) show enlarged views of individual particles outlined in gold for clarity. (**F**) Projection averages of the AKAP18γ–PKA$_{holo}$ complexes assembled with a truncated RIIα Δ44–86 construct using ISAC (**Yang et al., 2012**). (**G**) Projection averages of AKAP18γ–PKA$_{holo}$ complex assembled an RIIα ZeChimera construct. Scale bars in (**F**) and (**G**) represent 25 nm. (**H**) Statistical analysis of particle radius in angstroms (Å) for each AKAP18γ–PKA$_{holo}$ complexes. Box plot displays second and third quartile values, tails corresponding to minimum and maximum distances, (**) indicates $p < 0.01$; (****) indicates $p < 0.0001$.

In the second phase, experiments were conducted using higher-order complexes formed with wild-type RIIα, RIIα Δ44–86, or the RIIα ZeChimera (assembled as described above) to investigate whether manipulating the intrinsic flexibility of PKA altered phosphorylation of anchored PDE4D (*Figure 4F–G*). We measured cAMP-independent phosphorylation of PDE4D at 5 min, a time point that showed sub-maximal substrate phosphorylation (*Figure 4F,G*, *Figure 4—figure supplement 1*). Basal PDE4D phosphorylation was enhanced $1.97 \pm 0.18$-fold ($n = 6$, $p < 0.05$) in complexes formed with RIIα Δ44–86 when compared to a wild-type complex (*Figure 4G*, bars 1 and 4). In contrast, extension of the linker region in the context of AKAP18γ–RIIα ZeChimera PKA$_{holo}$ assembly had no effect as compared to wild type (*Figure 4G*, bars 1 and 7). Control experiments confirmed that addition of cAMP further augmented phosphorylation of PDE4D in all cases (*Figure 4F*, lanes 2, 5 and 8) and pretreatment with PKI inhibitor peptide abolished anchored kinase activity (*Figure 4F*, lanes 3, 6 and 9). These data show that the AKAP can be thought of as a catalyst that physically brings the reactants together, and the flexibility within the anchored PKA holoenzyme allows for the precise orientation of the enzyme and substrate. This mechanism may be particularly relevant for cAMP-independent phosphorylation events that are believed to represent approximately 30% of PKA action (*Taylor et al., 2012*). Therefore, this hitherto unexplained but critical component of cellular PKA activity may be accomplished by the persistent phosphorylation of substrates embedded in higher-order AKAP signaling assemblies.

To follow these in vitro studies, we tested our hypothesis that flexibility within the anchored PKA holoenzyme influences cAMP signaling in living cells. We generated a modified fluorescence resonance energy transfer (FRET) based PKA activity sensor using the A-kinase activity reporter (AKAR2) backbone (*Zhang et al., 2005*). Our modified sensor (AKAR-18$_{RBS}$) was constructed by fusing the PKA binding helix of AKAP18 (18$_{RBS}$) (*Fraser et al., 1998*; *Gray et al., 1998*) to the amino terminus of AKAR2 (*Figure 5A*). This genetically encoded reporter detects PKA phosphorylation in real-time by monitoring changes in the YFP/CFP emission ratio inside cells (*Figure 5A*). As a prelude to these studies AKAR-18$_{RBS}$ association with wild-type RIIα or either of the modified RIIα constructs was confirmed by co-immunoprecipitation of each complex from HEK293 cells (*Figure 5B*). In parallel, immunoblot and confocal fluorescent imaging analyses confirmed that mCherry tagged versions of each RIIα form were expressed to equivalent levels (*Figure 5C*) and uniformly distributed in HEK293 cells (*Figure 5—figure supplement 1A*).

Initial experiments evaluated basal FRET of the AKAR-18$_{RBS}$ reporter using a modified Leica DMI 6000B microscope. The raw YFP/CFP emission ratio of AKAR-18$_{RBS}$ was elevated in unstimulated cells expressing RIIα Δ44–86 as compared to RIIα wild type or the RIIα ZeChimera (*Figure 5D*). These data further develop the concept introduced in *Figure 4* that the RIIα linker region influences basal phosphorylation of AKAP-associated substrates. Next, real-time changes in the YFP/CFP emission ratio of the AKAR-18$_{RBS}$ reporter were monitored in cells expressing equivalent levels of the individual RIIα forms (*Figure 5E–G*). The β-adrenergic (β-AR) agonist isoproterenol (Iso) was administered after 100 s to initiate the cAMP response (*Figure 5E–G*). Notable increases in AKAR-18$_{RBS}$ FRET were evident in cells expressing RIIα wild type or the RIIα ZeChimera (*Figure 5E,G*). In both the cases the normalized FRET ratio was maximal at 200 s and gradually declined over the remainder of the time course (*Figure 5H*, black and orange traces). This latter phenomenon may be attributed to either desensitization of the β-AR system or dephosphorylation of the reporter by phosphoprotein phosphatases (*Bouvier et al., 1987*; *Pitcher et al., 1992*; *Violin et al., 2003*). Importantly, cells expressing the more compact

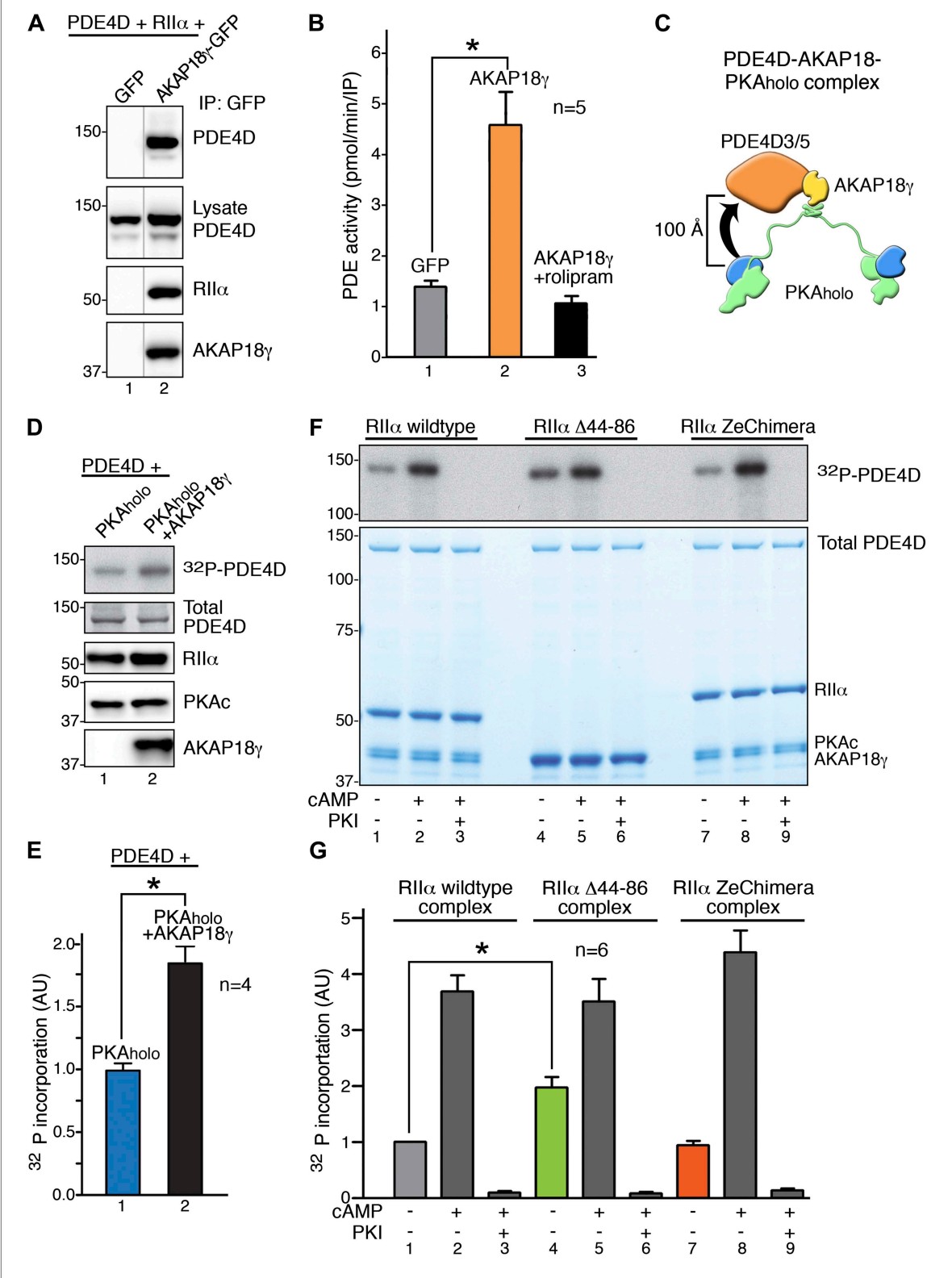

**Figure 4**. RIIα linker length influences basal PKA phosphorylation of associated substrates. (**A**) PDE4D isoforms associate with AKAP18 in cells. HEK293 cells expressing AKAP18–GFP or GFP alone along with PDE4D and RIIα were subjected to immunoprecipitation with anti-GFP antibodies. Immunocomplexes were separated by SDS-PAGE and immunoblotted for PDE4D and RIIα. AKAP18 was detected by RII overlay analysis. (**B**) Similar

*Figure 4. Continued on next page*

*Figure 4. Continued*

immunocomplexes as in (**A**) were used for phosphodiesterase activity assays. Inclusion of the small molecule rolipram (10 μM, bar 3) inhibited associated PDE4D activity. (**C**) Schematic of AKAP18–PKA holoenzyme-PDE4D3/5 complexes used in subsequent in vitro substrate phosphorylation assays. Based on our structural data, the PKA catalytic subunit is expected to be positioned within ~100 Å of its substrate PDE4D. (**D**) Anchoring of PKA and PDE4D stimulates cAMP-independent phosphorylation of the phosphodiesterase. (Top panel) Basal $^{32}$P incorporation into PDE4D (detected by autoradiograph) is shown in the absence or presence of AKAP18γ. (Bottom panels) Levels of PDE4D, RIIα, PKAc and AKAP18γ were assessed by immunoblot. (**E**) Densitometeric quantification of phospho-PDE4D in panel (**D**), n = 4 (p<0.05). (**F–G**) Deletion of the flexible linker augments cAMP-independent phosphorylation of PDE4D by 1.97 ± 0.18-fold (p<0.05). (**F**) (Top panel) Autoradiograph showing incorporation of $^{32}$P into PDE4D in each complex. (Bottom panel) Coomassie blue staining of the SDS-PAGE gel showing components of the assay. The PDE4D is at the top, while the different complexes are shown at the bottom. (**G**) Densitometric quantification of phospho-PDE4D levels in (**F**), n = 6.
The following figure supplements are available for figure 4:

**Figure supplement 1**. Phosphorylation of PDE4D is time-dependent.

RIIα Δ44–86 form responded more rapidly and robustly to isoproterenol with a maximal FRET response at 200 s (*Figure 5F,H*, green trace). The RIIα Δ44–86 effect was abolished when control experiments were performed with a proline modified AKAR-18$_{RBS}$ derivative that is unable to anchor PKA (*Fraser et al., 1998*; *Gray et al., 1998*; *Figure 5—figure supplement 1*). Other control experiments established that application of isoproterenol did not stimulate an AKAR-18$_{RBS}$ derivative (T/A) where the threonine phospho-acceptor was replaced with alanine (*Figure 5H*, inset, red trace).

Scatter plot representation of the Iso stimulated FRET responses at 200 s reveal that AKAR-18$_{RBS}$-PKA complexes formed with RIIα or the RIIα ZeChimera responded with similar magnitudes (*Figure 5I*, black and orange). In contrast, the normalized FRET responses of the more compact and less flexible AKAR-18$_{RBS}$-PKA complexes formed with RIIα Δ44–86 were significantly higher (*Figure 5I*, green, p<0.001). Taken together, these cell-based studies indicate that removal of the flexible linker region in RIIα augments cAMP-responsive PKA phosphorylation of AKAR-18$_{RBS}$ inside the cells. These findings complement the interpretation of our EM reconstructions and in vitro phosphorylation studies. These observations argue that compact and rigid AKAP–PKA assemblies enhance kinase action within the immediate vicinity of the substrate.

Local activation of PKA is believed to involve dissociation of the C subunits from the R subunit dimer. Yet, cAMP binds to the R subunits avidly (K$_D$ 6–8 nM), and its rate of release is so slow that it is not readily apparent how the cAMP-binding sites turnover within the cell (*Poppe et al., 2008*). Additionally, there is a report that cAMP can activate PKA without C subunit release (*Yang et al., 1995*). Therefore, we evaluated whether cAMP stimulation altered the composition of PKA holoenzymes associated with the AKAR-18$_{RBS}$ reporter. Interestingly, both the RII and C subunits of PKA were present in AKAR-18$_{RBS}$ immune complexes, even after stimulation of cAMP production by isoproterenol (*Figure 5J*, lanes 1 and 2). The isoproterenol-stimulated activation of PKA was validated by immunoblot detection of phosphorylated substrates with a phospho-PKA site antibody (*Figure 5J*, bottom panel). Control experiments confirmed that PKA subunits did not associate with the proline-modified AKAR-18$_{RBS}$ derivative, whereas the AKAR-18$_{RBS}$-T/A reporter retained the ability to anchor PKA (*Figure 5J*, lanes 3–6).

This AKAR-18$_{RBS}$ reporter data raised the question of whether anchoring can stabilize the PKA holoenzyme even in the presence of cAMP. To further test this hypothesis, co-precipitation experiments were repeated with full-length AKAP18γ. Again the C subunit was retained within the AKAP complex upon isoproterenol stimulation (*Figure 5K*, lanes 1 and 2). This occurred regardless of whether the PKA holoenzymes were formed with wild-type RIIα, RIIα Δ44–86, or RIIα ZeChimera (*Figure 5K*, top panel). These results confirm that the increased agonist-stimulated activity of the AKAR-18$_{RBS}$–RIIα Δ44–86 complex shown in *Figure 5H,I* is not due to enhanced dissociation of the PKA catalytic subunit. A broader implication is that release of the C subunit from anchored PKA holoenzymes may not be required for the phosphorylation of nearby substrates. Under this scenario, association with AKAPs limits and thereby defines the range of kinase action within the cell. This underscores the central role of AKAPs in the governance of cAMP responsive phosphorylation events.

A combination of steric effects and environmental factors may explain the enhanced catalytic efficiency of this condensed AKAP–PKA assembly inside cells. For example, the less flexible RIIα Δ44–86 dimer may constrain the PKAc subunits in a manner that minimizes the distance to their target

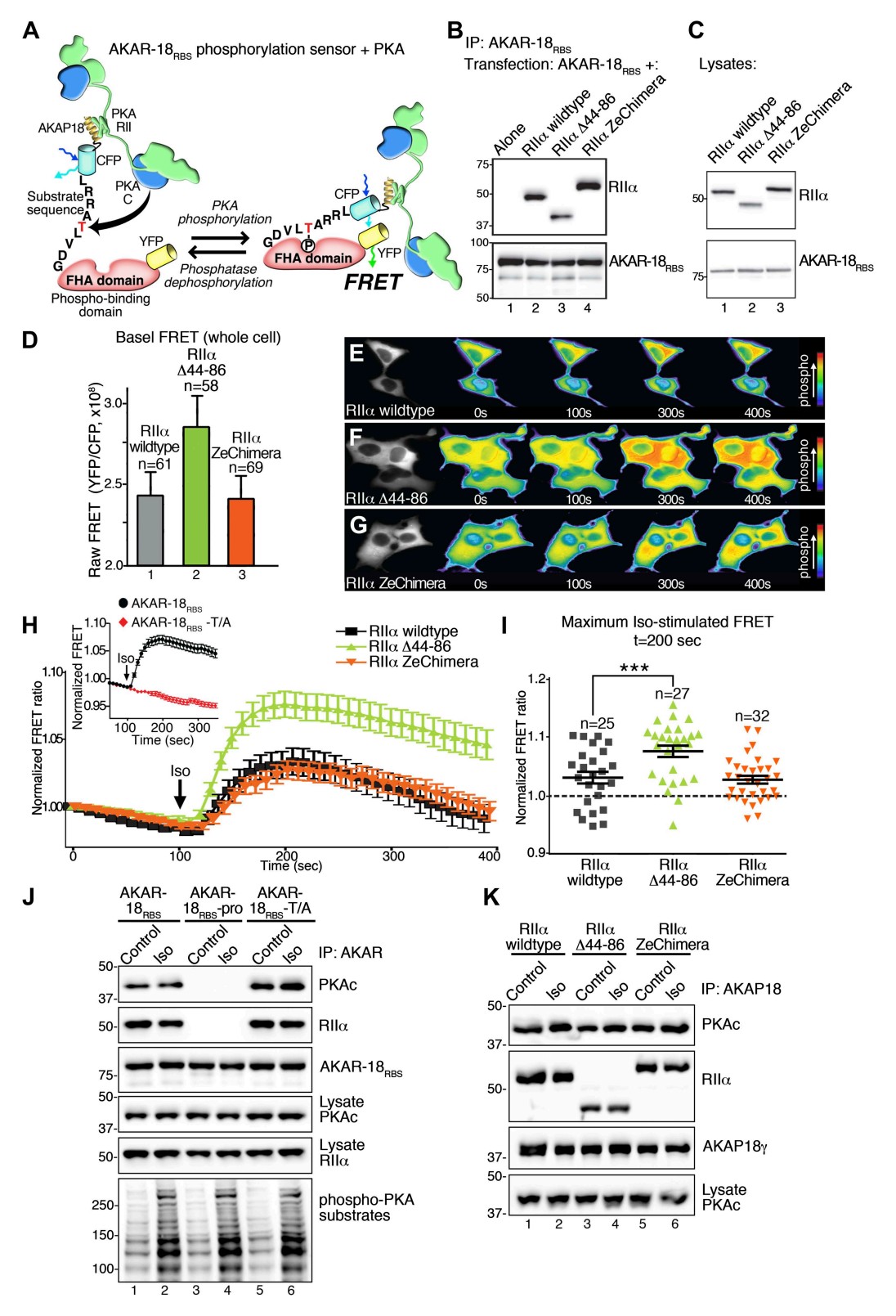

**Figure 5**. Flexibility within the anchored PKA holoenzyme impacts cAMP responsive signaling inside cells. (**A**) Schematic of the modified AKAR-18$_{RBS}$ FRET reporter used in these studies. The PKA RII binding site common to all AKAP18 isoforms was fused to the N-terminus of AKAR2 to create a FRET-based kinase activity sensor that anchors type II PKA holoenzyme. Phosphorylation of a consensus site Thr by PKA induces recruitment of the

*Figure 5. Continued on next page*

*Figure 5. Continued*

adjacent Forkhead homology-associated (FHA) phospho-Thr binding domain. Subsequent rearrangement brings together the ECFP and YFP (citrine) moieties to produce an increase in FRET signal as readout of kinase activity. (**B**) Co-immunoprecipitation of AKAR-18$_{RBS}$-PKA complexes with holoenzymes composed of three different RII forms. AKAR-18$_{RBS}$ was immunoprecipitated with anti-GFP antibodies and bound PKA subunits were detected by western blotting. (**C**) Lysates from cells transfected with the AKAR-18$_{RBS}$ reporter and each of the three RIIαforms were immunoblotted to confirm similar expression levels in each cohort. (**D**) Basal (unstimulated) raw FRET signals from cells expressing RIIα (gray), RIIα Δ44–86 (green) and RIIα ZeChimera (orange). (**E** and **G**) Time course of AKAR-18$_{RBS}$ activation in response to the β-adrenergic agonist isoproterenol in cells co-expressing the FRET reporter with (**E**) RIIα wild type, (**F**) RIIα Δ44–86, or (**G**) RIIα ZeChimera. Isoproterenol (Iso, 10 μM) was added at t = 100 s and FRET was recorded for 5 min post-stimulation. Warmer colors indicate increasing phosphorylation as shown in the pseudo-color scale. (**H**) Amalgamated traces of the Iso stimulated changes in FRET in each cohort (0–400 s). Data are normalized to unstimulated basal FRET level for each respective RIIα form. Changes in the AKAR-18$_{RBS}$ normalized FRET ratio are shown from cells expressing RIIα wild type (black), RIIα Δ44–86 (green) and RIIα ZeChimera (orange). Mutation of the phosphoacceptor threonine in the FRET reporter to create AKAR-18$_{RBS}$-T/A blocks phosphorylation and abolishes FRET in response to Iso treatment (Inset). (**I**) Scatter plot representation of peak FRET signals from all cells at 200 s. This plot indicates that AKAR-18$_{RBS}$-PKA complexes formed with RIIα wild type (black) or the RIIα ZeChimera (orange) had similar peak responses, while complexes formed with RIIα Δ44–86 displayed significantly greater peak FRET responses (p<0.001). This plot also shows that in all cases, some cells fail to respond entirely; these non-responders are included in the amalgamated data analysis presented in (**H**). (**J**) C subunit of PKA association with AKAR-18$_{RBS}$, AKAR-18$_{RBS}$–pro or AKAR-18$_{RBS}$–T/A complexes. AKAR-18$_{RBS}$ and AKAR-18$_{RBS}$–T/A co-precipitate RIIα and endogenous PKA catalytic subunit; AKAR-18$_{RBS}$–pro, a mutant that cannot bind RII subunits, fails to co-precipitate PKA complexes (top panels). Iso treatment (1 μM, 5 min) does not cause dissociation of PKAc from the AKAR-18$_{RBS}$ or AKAR-18$_{RBS}$–T/A complexes (top panel, lanes 1–2 and 5–6). Control immunoblots show equivalent levels of AKAR-18$_{RBS}$, AKAR-18$_{RBS}$–pro and AKAR-18$_{RBS}$–T/A in immunoprecipitates as well as RIIα and PKAc expression in cell lysates (middle panels). Immunoblotting for phospho-PKA substrates (R-X-X-pS/T motif) confirms that Iso treatment activates endogenous β-ARs and initiates downstream phosphorylation events (bottom panel). (**K**) Immunoprecipitation of full-length AKAP18 complexes following Iso treatment. Cells expressing AKAP18γ and RIIα variants were treated with vehicle or Iso (1 μM, 5 min) and AKAP18γ complexes were immunoprecipitated. Immunoblotting shows that Iso has no effect on the amount of PKA catalytic subunit in complexes formed with RIIα wild type, RIIα Δ44–86, or RIIα ZeChimera.

The following figure supplements are available for figure 5:

**Figure supplement 1**. Comparison of AKAR-18$_{RBS}$ and AKAR-18$_{RBS}$–pro anchoring and FRET controls.

substrate. Moreover, the more compact AKAP-PKAΔ44-86-substrate assembly may augment kinase activity by inhibiting the actions of signal termination enzymes. This could involve the steric exclusion of cellular protein phosphatases that catalyze the removal of phosphate or protection from phospho-diesterase activity that metabolizes cAMP. Deletion of the flexible linker between residues 44–86 of RIIα not only enhances basal phosphorylation of these anchored substrates, but also supplements the catalytic efficiency of the anchored PKA holoenzyme in this cellular context. However, this enhanced anchored kinase action may be an undesirable feature in vivo. Hence, we propose that the evolution of flexible linkers in RII may be a means to ensure bi-directional regulation of phosphorylation events by signal termination enzymes.

It is noteworthy that extending the number of residues in the RIIα ZeChimera linker had little effect on the overall structure of the AKAP-PKA complex, or on the activity of PKA monitored by our in vitro and cell-based assays. According to our structural analysis, this construct samples a range of radial conformations that are very similar to wild-type particles (*Figure 3H*). Moreover, this observation is consistent with theoretical analyses of random coiled chain models that demonstrate the average end-to-end length of random coil scales by the square root of the number of amino acids in the chain (average length ~ $N^{0.5}$, where $N$ is the number of residues) (*Flory, 1975*). This statement implies that the average length separating the ends of a random coil reaches a plateau, or maximum, that is only marginally altered upon the inclusion of more residues. Consequently extending the linker in the context of the RIIα ZeChimera may explain why this affords minimal benefit to the overall structure and dynamics of this seemingly more extended and malleable anchored enzyme complex. Hence, evolutionary constraint of the RIIα linker domain to 45–47 residues may ensure appropriate separation between the anchoring and transduction domains of anchored mammalian PKA holoenzymes.

Another key element of our model is that the array of conformations adopted by the anchored RII subunit dimer defines a 'radius of action' for the tethered catalytic subunit. Not only does this structural plasticity explain why an intact PKA holoenzyme structure has eluded X-ray crystallographers for over 20 years but also suggests that the 'built in' flexibility within this protein complex sets a dynamic range for basal and cAMP-dependent phosphorylation of nearby substrates. This latter concept could have profound implications for PKA and other kinases, where enzyme activity is constrained within anchored complexes or enzyme scaffolds. The conformational variability of the tethered complex could

enhance kinase access to multiple phosphorylation sites on the same substrate, or, alternatively, orient each catalytic subunit toward distinct substrate proteins within the vicinity of the scaffolding site. Our structural and cellular characterization of higher-order AKAP assemblies emphasizes how static protein–protein interactions act in concert with the dynamic movement of sub-structures to orchestrate local phosphorylation events. We believe that these combined molecular approaches are broadly applicable for the investigation of related enzyme complexes that are spatially organized within the cell.

## Materials and methods

### Protein expression and purification

RIIα and PKAc were expressed in BL21(DE3)pLysS. *Escherichia coli* was transformed with pET28b-RIIα-6xHis (encoding full-length RIIα, accession NM_008924) or pET15a-6xHis-mPKAc (Addgene plasmid 14921; www.addgene.org), encoding full-length mouse PKA catalytic subunit (*Narayana et al., 1997*) and grown to an $OD_{600}$ of 0.6. Protein expression was induced with 1 mM IPTG, and cells were grown with shaking for 16 hr at 28°C (for RIIα-His) or 24 hr at 16°C (for PKAc). The cells were harvested by centrifugation at 5,000×*g* for 10 min at 4°C. Pellets containing induced protein were stored at −20°C until use. For purification, the cells were thawed in 50 mM NaPhosphate, pH 7.5, 100 mM NaCl, 10 mM Imidazole, 2 mM $MgCl_2$ containing 4 mg/ml lysozyme, 1 mM AEBSF, 50 µg/ml soybean trypsin inhibitor, 2 µg/ml leupeptin, 16 µg/ml benzamidine, and 25 U/ml benzonase. After resuspension, Triton X-100 was added to 0.5% and lysates were rocked at 4°C for 30–60 min. When the lysate was no longer viscous, debris was pelleted by centrifugation at 37,000×*g* for 30 min at 4°C. Cleared lysates were added to 1/25 vol $Ni^+$–NTA beads (bed volume; GE Healthcare, Pittsburg, PA) and rocked for 1 hr at 4°C. The bead slurry was then added to a column and allowed to settle. Beads were washed with >10 bed volumes of 50 mM NaPhosphate, pH 7.5, 500 mM NaCl, 20 mM Imidazole. Purified protein was eluted by collecting 2 × 2 bed volume fractions of 50 mM NaPhosphate, pH 7.5, 500 mM NaCl, 50 mM Imidazole followed by 4 × 2 bed volumes 50 mM NaPhosphate, pH 7.5, 500 mM NaCl, 300 mM Imidazole. Fractions were analyzed by SDS-PAGE and Coomassie staining. The fractions containing the protein of interest were pooled, concentrated and applied to a Superdex 16/600 S200 gel filtration (GF) column. The column was run in 25 mM HEPES, pH 7.5, 200 mM NaCl, 1 mM EDTA, 1 mM TCEP. Peak fractions were pooled, concentrated, brought to 10% glycerol and frozen in liquid $N_2$ for storage at −80°C.

AKAP18γ (accession NM_016377 [Note: This sequence was updated during the course of these studies and corresponds to the longer AKAP18δ isoform; the construct used in these studies encodes the γ isoform starting at Met23 of the referenced ORF]) was expressed with an N-terminal strep-II tag and a C-terminal 6x-His tag in High Five insect cells (*Trichopulsia ni*; Invitrogen). Sf-9 cells were transfected with a Bacmid containing the expression cassette from pFastBac-C-His-strep-AKAP18γ. After two rounds of viral amplification, High Five cells were infected with baculovirus (at a cell density of ~2 × $10^6$ cells/ml) and grown for 72 hr at 26°C shaking at 105 rpm. Cells were harvested by centrifugation (10 min, 700×*g*) and frozen at −20°C. Protein was purified as above using $Ni^+$–NTA Sepharose, except lysozyme was omitted from the lysis buffer. Gel filtration chromatography was performed as described above.

MBP-PDE4D3-8xHis was expressed in BL21(DE3)pLysS transformed with the vector pMAL-C5P2-PDE4D3-8His, which was constructed by PCR and InFusion (Clontech) cloning into a modified pMAL vector (NEB) containing an 8xHis tag prior to MBP and a PreScission protease cleavage site after MBP. Protein was purified using $Ni^+$–NTA Sepharose, and gel filtration chromatography was performed as described above.

### Molecular biology, cloning, and mutagenesis

Constructs for deleted and extended linker forms of mRIIα were created using standard cloning methods. Briefly, to create the linker deletion, AscI restriction sites were introduced into the cDNA by site-directed mutagenesis. The resulting plasmid was digested with AscI and then re-ligated closed. To create the Zebrafish linker chimera, a minigene encoding the Zebrafish linker from NCBI Reference sequence NM_212958.1 was synthesized with flanking AscI sites (Integrated DNA Technologies). This minigene was digested out of the carrier vector and inserted into the AscI sites in the modified RIIα cDNA. Clones were screened for orientation and verified by sequencing (Genewiz). Mammalian expression constructs encoding RII variants were created by PCR cloning into Gateway DONR vectors (Invitrogen) followed by LR cloning into pcDNA-DEST40. Vectors that express AKAR-18$_{RBS}$, the AKAR-18$_{RBS}$–T/A and the AKAR-18$_{RBS}$–pro (L9P, A13P in the AKAP18 anchoring helix) mutants were

created upon insertion of a cDNA encoding the PKA binding site from AKAP18 into pcDNA3-AKAR2.2 (*Zhang et al., 2005*) followed by site directed mutagenesis (QuikChange XL II, Stratagene).

## AKAP complex formation

AKAP18γ–PKA complexes were assembled using purified proteins. RIIα and PKAc were mixed at a 1:1.2 molar ratio, and AKAP18γ was added in molar excess to favor the isolation of stoichiometric complexes by gel filtration. Complexes were dialyzed against 15 mM MOPS, pH 6.5, 100 mM NaCl, 1 mM $MgCl_2$, 1 mM TCEP, 2% glycerol for >4 hr at 4°C in 0.5 ml Slide-A-Lyzer MINI dialysis cups (88401; Thermo Scientific). The complexes were then loaded onto a Superdex 16/600 S200 gel filtration column on an AKTA Purifier (GE Healthcare) and separated at a flow rate of 0.5 ml/min in 25 mM HEPES, pH 7.5, 200 mM NaCl, 1 mM EDTA, 1 mM TCEP. Peak fractions were analyzed by SDS-PAGE and Coomassie staining. The fractions containing approximately stoichiometric complex components were pooled, concentrated, brought to 10% glycerol and frozen in liquid $N_2$ for storage at −80°C. RIIα-PKAc holoenzyme complexes were assembled as above, omitting AKAP18γ.

## Western blotting

Duplicate samples were separated on 10% Tris-Glycine-SDS gels. Proteins were transferred to nitrocellulose membranes and blocked with 5% NFDM, 1% BSA in TBST. Membranes were probed with either anti-AKAP18 rabbit polyclonal antiserum (VO57), mouse anti-PKAc or anti-RIIα monoclonal antibodies (BD Biosciences) overnight at 4°C. The membranes were washed, incubated with appropriate secondary antibodies, washed again and protein was detected using ECL (Super-signal Pico, Thermo Pierce) on an AlphaInnotech Multiimager III.

## Native PAGE

AKAP18γ–PKA complexes were analyzed by native PAGE using precast 3–8% Tris-Acetate gels (Invitrogen). The samples were mixed with Invitrogen's 2X native sample buffer and run in 50 mM Tris pH 8.3, 50 mM Tricine, along with native MW markers (Invitrogen). Protein was detected by silver staining (SilverQuest, Invitrogen).

## Phosphodiesterase assays

PDE assays were performed as described (*Hill et al., 2005*). Briefly, after washing, immunocomplexes were suspended in 50 µl KHEM (50 mM KCl, 50 mM HEPES-KOH, pH 7.2, 10 mM EGTA, 1.92 mM $MgCl_2$). Next, 50 µl of PDE assay buffer (20 mM Tris-HCl, pH 7.4, 10 mM $MgCl_2$, 2 µM cAMP, 100 nM okadaic acid) containing ~50,000 cpm [$^3$H]-cAMP was added, and the samples were incubated at 30°C for 20 min with shaking. The samples were boiled for 3 min and incubated on ice for 15 min. Snake venom (25 µl of a 1 mg/ml solution) was added, and the samples were incubated at 30°C for 10 min with shaking. Dowex AG 1×8 (200–400 mesh, CL form, washed and used as a 1:1:1 slurry of resin:water:ethanol) ion exchange resin (400 µl) was added, and the samples were incubated on ice for 15 min. The resin was centrifuged for 3 min at 14,000×$g$, and 150 µl of the supernatant was mixed with 1 ml scintillation fluid for counting. All assays were performed in duplicate.

## Kinase assays

PKA holoenzyme was incubated in the absence or presence of molar equivalent of purified AKAP18γ. These pre-formed complexes (1 µg) were mixed with excess substrate MBP-PDE4D3 (2 µg) in 40 mM Tris, pH 7.5, 10 mM MgOAc, 0.1 mM EGTA, 100 µM IBMX. $Mg^{2+}$-ATP containing 1 µCi $^{32}$P-ATP was added to a final concentration of 10 µM. The samples were incubated at 30°C with rapid shaking for 5 min. Reactions were stopped by the addition of 5X Laemmli sample buffer and boiling for 5 min. The samples were separated by SDS-PAGE, and gels were either stained with Coomassie blue or transferred to nitrocellulose for India ink staining. Phosphorylation of PDE4D3 was detected by auto-radiography. Films were scanned for quantification by densitometry, followed by analysis using GraphPad Prism. Analysis of AKAP18γ–PKA complexes containing different RIIα variants was performed and analyzed similarly. In these assays, PKI was used at a final concentration of 10 µM and added to appropriate samples 10 min prior to the start of the assay. Prior to initiation of the assay, cAMP was added to some samples to a final concentration of 30 µM.

## FRET measurements

HEK293 cells were transiently transfected with cDNAs encoding AKAR-18$_{RBS}$ and individual RII subunits, and cultured on 35-mm glass coverslip dishes (MatTek Corporation). 48 hr after transfection,

the cells were imaged in HEPES-Buffered Saline Solution (HBSS; 116 mM NaCl, 20 mM HEPES, 11 mM Glucose, 5 mM NaHCO$_3$, 4.7 mM KCl, 2.5 mM CaCl$_2$, 1.2 mM MgSO$_4$, 1.2 mM KH$_2$PO$_4$, pH 7.4). The cells were stimulated by isoproterenol (10 µM). Fluorescence emission was acquired using a DMI6000B inverted microscope (Leica), an EL6000 component (fluorescent light source, filter wheel, ultra fast shutter, Leica) and a CoolSnap HQ camera (Photometrics), all controlled by MetaMorph 7.6.4 (Molecular Devices). Dual-emission images were obtained simultaneously through a Dual-View image splitter (Photometrics) with S470/30 and S535/30 emission filters and 505 dcxr dichroic mirror (Chroma). Exposure time was 200 ms with an image interval of 10 s. FRET changes within a region of interest were calculated as the ratio of measured fluorescent intensities from two channels (M$_{donor}$, M$_{indirectAcceptor}$) after subtraction of background signal. FRET ratio (YFP/CFP) changes were normalized to the average FRET ratio value before stimulation.

## Immunoprecipitations and immunoblotting

HEK293 cells were transiently transfected (TransIT LT1; Mirus) with vectors encoding various proteins according to figure legends. After 40–48 hr and prior to harvesting, the cells were serum starved in DMEM for 1 hr at 37°C. The cells were then treated with vehicle or Isoproterenol for 5 min at 37°C and harvested in lysis buffer (25 mM HEPES, pH 7.4, 150 mM NaCl, 1 mM EDTA, 1 mM EGTA, 20 mM NaF, 2% glycerol, 0.3% Triton X-100) containing protease inhibitors. AKAP18γ or AKAR-18$_{RBS}$ complexes were immunoprecipitated with anti-GFP rabbit IgG (Invitrogen) and protein A agarose for 2 hr at 4°C. Beads were washed 1 × 1 ml in lysis buffer containing 350 mM NaCl, 1 × 1 ml wash buffer (lysis buffer with no detergent) containing 350 mM NaCl and 2 × 1 ml wash buffer. Proteins were separated on 4–12% gradient gels (Invitrogen) and transferred to nitrocellulose membranes. Primary antibodies (PKA catalytic mAb [BD Biosciences], 1:1000; RIIα mAb [BD Biosciences], 1:2000; GFP mAb [Santa Cruz Biotechnology], 1:2000; phospho-PKA substrates rabbit mAb [Cell Signaling Technology], 1:1000) were incubated with membranes overnight at 4°C in TBST/Blotto. The membranes were washed extensively in TBST, incubated with secondary antibodies, washed again and developed using ECL (Thermo Pierce) on an Alpha Innotech MultiImage III with FluoroChem Q software.

## Confocal imaging

Confocal microscopy was performed using a Zeiss LSM-510 META laser scanning confocal microscope equipped with a Zeiss 63 × g, 1.4 numerical aperture, oil immersion lens. For live cell imaging, HEK293 cells expressing AKAR-18$_{RBS}$ and individual RII-mCherry constructs were imaged in HBSS. Colocalization studies were performed using dual excitation (488, 543 nm) and emission (bandpass 505–530 nm and long-pass 560 nm for YFP and mCherry, respectively) filter sets.

## Electron microscopy

The wild-type and mutant AKAP18γ–PKA holoenzyme complexes were prepared similarly for electron microcopy studies. Freshly purified samples were diluted 1:25 times to a final concentration of 5 µg/ml with EM buffer containing 25 mM HEPES (pH—7.4), 200 mM NaCl, 0.5 mM EDTA and 1 mM dithiothreitol (DTT). For gold-labeling studies, a 0.05% solution of streptavidin-conjugated gold particles (5 nm, Nanocs) was diluted 1:100 with EM buffer and mixed 1:1 (vol/vol) with the wild-type AKAP18γ–PKA complex and allowed to incubate for 1 hr at 4°C. 2 µl of specimen samples were applied to carbon coated EM grids and negatively stained with 0.75% (wt/vol) uranylformate. Wild-type particles were visualized on a 120 kV TEM (FEI), and images were recorded at a nominal magnification of 52,000 × at the specimen level on a 4 k × 4 k CCD (Gatan). Serial tomographic images were obtained using automated data collection software (Xplore3D, FEI) collected at tilt angles of 0°, 20°, 40° and 50° (*Figure 2—figure supplement 1*). The mutant complexes and gold-labeled particles were visualized on a 120 kV TEM (FEI) and recorded at a nominal magnification of 68,000 × at the specimen level on a 4 k × 4 k CMOS-based camera (Tietz).

For image processing, micrographs were binned two times yielding final pixel sizes of 4.2 Å per pixel and 3.0 Å per pixel for the wild-type dataset and mutant datasets, respectively. Thon rings in the power spectra were used to select only those micrographs free of drift or significant astigmatism. The contrast transfer function (CTF) parameters were determined for each micrograph using the program CTFTILT (*Mindell and Grigorieff, 2003*). ~7,000 wild-type particles and 1,000 mutant particles were hand selected from micrographs using Ximdisp (*Smith, 1999*). Two-dimensional (2D) projection averages were generated using reference-free multivariate statistical analysis methods in IMAGIC

(*van Heel et al., 1996*) and with the Iterative Stable Alignment and Clustering routines implemented using the ISAC program (*Yang et al., 2012*). Initial three-dimensional (3D) reconstructions were generated from 0°, 20°, 40° and 50° tilted particle class-average datasets displaying either linear or bent conformations using angular reconstitution routines in IMAGIC (*van Heel et al., 1996*). Refinement of the initial 3D density maps was performed using individual particle images in FREALIGN (*Grigorieff, 2007*) without symmetry constraints. The final 3D density maps were filtered to 35 Å resolution as suggested by Fourier shell correlation (FSC) analysis. Back-projection analysis was carried out in SPIDER (*Frank et al., 1996*). Statistical analysis of individual particle dimensions was obtained by measuring particle lengths on unbinned micrographs using ImageJ (*Schneider et al., 2012*) and evaluated by the two-tailed $t$ test.

### Modeling of the AKAP18γ–PKA holoenzyme complex

Structural models of the AKAP18γ–PKA holoenzyme complex were constructed to represent the triangular and extended conformations determined by the 3D reconstructions. A model of AKAP18γ (residues 88–317) was initially constructed as follows. The atomic coordinates of the AKAP–PKA binding helix in complex with the RIIα docking and dimerization domain ($RII_{D/D}$-$AKAP_{helix}$; PDB 2IZX) (*Gold et al., 2006*) was used as a template to model the corresponding region in AKAP18γ, residues 301–317, by computational mutation and minimization in COOT (*Emsley and Cowtan, 2004*). The $AKAP_{helix}$ domain was oriented with the N-terminus (residue 301) placed in close proximity to the C-terminus of the AKAP18γ central domain (PDB 2VFL) (*Gold et al., 2008*), residues 88–290. The relative orientations were guided by their corresponding fit into the 3D density map. The two AKAP18γ domains were connected by a 9-residue linker corresponding to the AKAP18γ sequence 291–299 using COOT (*Emsley and Cowtan, 2004*). This model of the linked AKAP18γ structure (residues 88–317) bound to the $RII_{D/D}$ domain was placed within the central density of the density map, and the fit was refined by computational minimization in Chimera (*Pettersen et al., 2004*). The central density was determined to be the site of AKAP18γ localization by affinity gold labeling studies (see above).

The atomic coordinates corresponding to the mouse PKA catalytic domain (PKAc) bound to the RIIα cAMP binding domain (PDB 2QVS) (*Wu et al., 2007*) were placed into each of the peripheral EM densities, and their fits were independently minimized in Chimera (*Pettersen et al., 2004*). Each RIIα regulatory domain (residues 91–392) was connected to a corresponding $RII_{D/D}$ domain (residues 3–43) by a 46-residue linker region (corresponding to residues 44–90 from mouse) using the modeling program COOT (*Emsley and Cowtan, 2004*). The RIIα linker regions were modeled as unstructured polypeptides connecting the sub-domains of the AKAP18γ-PKA holoenzyme complex oriented in the triangular and linear conformations determined by EM.

Figures were prepared in Adobe Photoshop. Protein structures were visualized and images captured in UCSF chimera version 1.6.

## Acknowledgements

The authors would like to thank Nikolaus Grigorieff (Janelia Farm Research Campus, Howard Hughes Medical Institute) for critically reading this manuscript, as well as Alexis Rohou, Don Olbris, Mark Bolstad, and Sean Murphy (Janelia Farm Research Campus, Howard Hughes Medical Institute) and Katherine Forbush (University of Washington) for technical assistance. The authors also thank Patrick J Nygren (University of Washington) for valuable discussions and insight.

EM maps have been deposited in the EM Data Bank. Atomic coordinates of composite models for the AKAP18γ-PKA$_{holo}$ complex have been deposited in the Protein Data Bank.

## Additional information

### Funding

| Funder | Grant reference number | Author |
| --- | --- | --- |
| Howard Hughes Medical Institute | | F Donelson Smith, Steve L Reichow, Jessica L Esseltine, Dan Shi, Lorene K Langeberg, John D Scott, Tamir Gonen |

| Funder | Grant reference number | Author |
|---|---|---|
| National Institutes of Health | HL088366, GM48231 | F Donelson Smith, Jessica L Esseltine, Lorene K Langeberg, John D Scott |

The funders had no role in study design, data collection and interpretation, or the decision to submit the work for publication.

## Author contributions

FDS, SLR, JLE, Conception and design, Acquisition of data, Analysis and interpretation of data, Drafting or revising the article; DS, Conception and design, Acquisition of data; LKL, Conception and design, Drafting or revising the article; JDS, TG, Conception and design, Analysis and interpretation of data, Drafting or revising the article

# Additional files

## Major dataset

### The following datasets were generated:

| Author(s) | Year | Dataset title | Dataset ID and/or URL | Database, license, and accessibility information |
|---|---|---|---|---|
| Smith FD, Reichow SL, Esseltine JL, Shi D, Langeberg LK, Scott JD, Gonen T | 2013 | 3D EM reconstruction of the AKAP18-PKA complex in a bent conformation. (corresponding with PDB 3J4Q) | EMD-5755; http://www.ebi.ac.uk/pdbe/entry/EMD-5755 | Publicly Available at Electron Microscopy Data Bank (EMDB). |
| Smith FD, Reichow SL, Esseltine JL, Shi D, Langeberg LK, Scott JD, Gonen T | 2013 | 3D EM reconstruction of the AKAP18-PKA complex in a linear conformation. (corresponding with PDB 3J4R) | EMD-5756; http://www.ebi.ac.uk/pdbe/entry/EMD-5756 | Publicly Available at Electron Microscopy Data Bank (EMDB). |
| Smith FD, Reichow SL, Esseltine JL, Shi D, Langeberg LK, Scott JD, Gonen T | 2013 | Pseudo-atomic model of the AKAP18-PKA complex in a bent conformation derived from electron microscopy | 3J4Q; http://www.rcsb.org/pdb/explore/explore.do?structureId=3J4Q | Publicly Available at Protein Data Bank (PDB). |
| Smith FD, Reichow SL, Esseltine JL, Shi D, Langeberg LK, Scott JD, Gonen T | 2013 | Pseudo-atomic model of the AKAP18-PKA complex in a linear conformation derived from electron microscopy | 3J4R; http://www.rcsb.org/pdb/explore/explore.do?structureId=3J4R | Publicly Available at Protein Data Bank (PDB). |

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
