## [Decision Letter]

Thank you for sending your work entitled “Optimal phosphorylation by anchored kinase activity is mediated by intrinsic disorder” for consideration at *eLife*. Your article has been evaluated by a Senior editor and 3 reviewers, one of whom is a member of our Board of Reviewing Editors.

The Reviewing editor and the other reviewers have discussed their comments and the Reviewing editor has assembled the following comments. Please respond to these comments by email to let us know how you intend to prepare a revised manuscript, rather than actually preparing a revised manuscript at this stage.

AKAP proteins play an important role in anchoring PKA and other kinases allowing targeted regulatory phosphorylation. There has been rather little structural information on AKAP complexes. New structural data has the potential to provide significant new insights into the mechanisms of this and related systems. Here, the authors have purified a complex corresponding to a single AKAP18γ molecule together with two copies of the PKA RIIα subunit and two PKAc subunits. The complex was subjected to negative stain electron microscopy and single particle analysis. The authors identify a tripartite architecture for the complex with the AKAP protein located centrally in close association with two dimerization interface domains of the RII subunits. Conversely, the two PKAc subunits closely associated with the cAMP binding domains of the RII subunits are mapped to the peripheral densities. The authors provide biochemical data to support the conclusion that the architecture of the AKAP complex contributes to PKA-mediated phosphorylation dynamics.

The present analysis of the electron microscope data should be expanded and improved to justify the authors' conclusions (points 1–2). There are also concerns about the biochemical analysis presented (points 3–5).

1) A major conclusion of the paper is that the complex is highly flexible, adopting conformations which range from triangular to linear configurations and that this flexibility serves to allow catalytic interaction with a range of potential substrates. The concept of a highly flexible complex is initially introduced in interpreting the appearance of raw images and class averages of the complex (Figure 1). However, variations in appearance of such projection images can in principle arise from either conformational variability or from projections of the three-dimensional structure in different directions. The 2D analysis of the projection images does to not appear to be optimal. The class averages in Figure 1 appear to be over-averaged with quite low levels of detail – in comparison there appear to be significant details in the original images that are not recovered in the class averages. A more powerful classification procedure such as the ISAC program developed in the Penzcek lab may result in higher-quality class averages and aid the analysis.

2) The authors use the angular reconstitution approach to calculate 3D structures for the complex in extended and linear conformations using tilted images. This provides stronger evidence for the existence of multiple conformations. However, there is a significant concern here with respect to the analysis of tilted negatively stained images. Although the negative stain approach yields relatively faithful images at zero tilt, it is typically associated with substantial shrinkage normal to the surface of the grid. Hence tilted images are commonly distorted from their proper projection structures. This could lead, for example, to the apparent linearization of a triangular conformation. Careful analysis of untilted images exploiting the most effective classification approaches should make it possible to recover a sufficient range of projections from untilted images for 3D analysis.

3) The in vitro phosphorylation studies imply that the major role of RIIα linker is on basal (not PKA-stimulated) phosphorylation (Figure 4). However, the in vivo reporter analysis (Figure 5) does not appear to show changes in basal activity, but does show changes in stimulated activity. How do the authors reconcile these findings? These observations require some explanation because they lie at the heart of the authors' conclusions.

4) Controls should be included in the in vivo analysis (Figure 5). What is the effect of blocking AKAP/RIIα interactions (e.g., using peptide competition or by using a mutant construct that does not bind RIIα) on the FRET signal for each of the three RIIα proteins?

5) There are potential alternative explanations for the biochemical observations presented that are unrelated to substrate tethering and orientation that need to be excluded by experimental analysis. For example, the changed activity reported upon alteration of the linker could involve more rapid or complete dissociation of PKAc from the complex. The authors should therefore examine levels of associated PKAc in immunoprecipitates of the various RII constructs prior to, and subsequent to, a cAMP agonist.

---

## [Author Response]

*1) Variations in appearance of such projection images can in principle arise from either conformational variability or from projections of the three-dimensional structure in different directions*.

We agree with the reviewers that the appearance of structural variation can result from either conformational variability or from projections of the structure in different orientations. We have expanded our EM analysis to include a new supporting figure showing single particles at 0-degree tilt that appear as either linear or triangular conformations, together with a montage of the same particles collected under various tilt-angles. This analysis clearly shows that the linear and triangular particles are distinct and do not appear as one another at the various angles that we sample. These data are presented in Figure 2—figure supplement 1.

Moreover, we have included back-projection analysis of our 3D-reconstructions (both linear and triangular). This analysis shows that even when the triangular model is tilted-completely on its edge (where it may appear more linear in projection) its dimensions are always smaller than the extended end-to-end particle lengths obtained for the truly linear particles. These data has been added to Figure 2—figure supplement 2.

These additional analyses suggested by the reviewer both improve and support our assessment that the structural variability observed by EM is a result of the inherent conformational variability within the AKAP-PKA complex.

*A more powerful classification procedure such as the ISAC program developed in the Penzcek lab may result in higher quality class averages and aid the analysis*.

As suggested by the reviewers we have used ISAC in an attempt to tease out more details in our projection averages. These new and improved images are included in Figure 1, Figure 3, and Video 1.

*2) Although the negative stain approach yields relatively faithful images at zero tilt, it is typically associated with substantial shrinkage normal to the surface of the grid. Hence tilted images are commonly distorted from their proper projection structures. This could lead, for example, to the apparent linearization of a triangular conformation*.

The reviewers raise an important concern regarding the effects of particle flattening. The EM analysis described above has addressed this concern. We have modified the text to more clearly reflect the concern regarding flattening artifacts in negative stain EM. Of course one could consider cryo-EM as an alternative; however, these particles are too small and the contrast will be too weak. Consequently this is not a viable option. But we also would like to note that our analysis represents the best interpretation of the available data and structural findings are corroborated with biochemical experiments.

*3) The in vitro phosphorylation studies imply that the major role of RIIα linker is on basal (not PKA-stimulated) phosphorylation (*Figure 4*). However, the in vivo reporter analysis (*Figure 5*) does not appear to show changes in basal activity, but does show changes in stimulated activity. How do the authors reconcile these findings? These observations require some explanation because they lie at the heart of the authors' conclusions*.

We are happy to include additional data showing the basal activity of each modified PKA holoenzyme using our FRET sensor. Analysis of raw baseline FRET activity shows elevated signal of the truncated RII form as compared to the wild type or the extended RII chimera. This additional data further supports the concept that the linker region within the anchored PKA holoenzyme is important for establishing spatial constraints for substrate phosphorylation under conditions where sustained basal phosphorylation is necessary to maintain homeostasis. These data are now included in Figure 5 and discussed in the text.

In response to the second aspect of this question we are happy to clarify that nature of the dynamic changes in the cAMP responsive FRET ratios shown in Figure 5. These data represent the normalized rate of change in the CFP/YFP ratio that occurs upon cAMP stimulation. Accordingly, these data measure hormone stimulated reporter activity over and above the raw baseline FRET. We have clarified this point in the figure legends.

We maintain that the changes in basal and stimulated activity are both important indications that linker length and “disorder” are crucial to maintain an appropriate amount of built-in flexibility and allow the kinase to phosphorylate target substrates in a spatially constrained microenvironment.

*4) Controls should be included in the in vivo analysis (*Figure 5*). What is the effect of blocking AKAP/RIIα interactions (e.g., using peptide competition or by using a mutant construct that does not bind RIIα) on the FRET signal for each of the three RIIα proteins*?

We now include supplemental data that monitors the responsiveness of an AKAP18_RBS_-pro-AKAR containing two proline substitutions in the AKAP18 anchoring helix that abolish binding to PKA. Experiments using this control kinase activity reporter were performed to analyze all three PKA holoenzyme forms (Figure 5—figure supplement 1). These data are discussed in the text.

*5) The authors should therefore examine levels of associated PKAc in immunoprecipitates of the various RII constructs prior to, and subsequent to, a cAMP agonist*.

This is an interesting point. To test this concept, cells were treated with isoproterenol and AKAP18 complexes were immunoprecipitated. The amount of associated PKA catalytic subunit was determined by western blotting. Interestingly, very little PKA C subunit dissociates from any of the complexes, and there is not a noticeable difference between PKA holoenzymes assembled or reconstituted with each of the different RII forms.

This new and exciting data, included in Figure 5, suggest that anchoring may function to stabilize PKA holoenzymes in a manner that prevents full dissociation of the subunits upon occupancy of the cAMP binding sites. These results are discussed in the text.